Comparative leaf anatomy of two species of Ipomoea L. (Convolvulaceae): taxonomic importance and adaptations to xeric conditions of the cangas

Silva Joyce C. R. 1
Silva Kleber R. 1
http://orcid.org/0000-0003-4762-3515 Caldeira Cecilio F. 1
Oriani Aline 2
http://orcid.org/0000-0001-9690-5565 Watanabe Mauricio T. C. 1 mauricio.watanabe@itv.org
1 Instituto Tecnológico Vale , Belém, Pará , Brazil
2 Escola Superior de Agricultura Luiz de Queiroz, Universidade de São Paulo , Piracicaba, São Paulo , Brazil
De Costa Janendra
Electronic publication date: 2024 Dec 10
Publication date: 2024
Volume: 12
Electronic Location ID: e18599
Received 2024 Sep 10; Accepted 2024 Nov 6
Copyright: © 2024 Silva et al.
Copyright year: 2024
Copyright holder: Silva et al.
License: This is an open access article distributed under the terms of the Creative Commons Attribution License, which permits unrestricted use, distribution, reproduction and adaptation in any medium and for any purpose provided that it is properly attributed. For attribution, the original author(s), title, publication source (PeerJ) and either DOI or URL of the article must be cited.
License URL: https://creativecommons.org/licenses/by/4.0/

Keywords: Amazon rainforest, Ipomoea cavalcantei, Ipomoea marabaensis, Morning glories, Morphology, Taxonomy, Environmental adaptations, Rocky fields

Funding: Conselho Nacional de Desenvolvimento Científico e Tecnológico (CNPq) 402770/2018-8 This study was funded by the Conselho Nacional de Desenvolvimento Científico e Tecnológico (CNPq) (research funding 402770/2018-8). The funders had no role in study design, data collection and analysis, decision to publish, or preparation of the manuscript.

==============================
Background

Accurate species identification is the first step towards establishing conservation strategies, especially regarding rare and threatened species, such as those studied here. Moreover, understanding the responses to the environment and growing conditions of endemic species is necessary for its conservation. This study compares the leaf anatomy of Ipomoea cavalcantei and I. marabaensis, which grow on the Brazilian ironstone outcrops (cangas) and exhibit morphological convergence.

Methods

Leaf anatomical characters of the species were described. Additionally, the leaf adaptive potential of I. cavalcantei was evaluated, comparing individuals grown in natural canga areas (in situ) and cultivated in topsoil (ex situ). Quantitative analysis evaluated stomatal density, polar and equatorial diameter of stomata, and thickness of the epidermis and mesophyll.

Results

Ipomoea cavalcantei and I. marabaensis can be distinguished by the mesophyll type. Ipomoea marabaensis is also characterized by the presence of lateral protuberances on the abaxial surface of the midrib. Individuals of I. cavalcantei under cultivation have lower stomatal density, although their functionality (polar/equatorial diameter) is more significant than those grown in a natural environment; these individuals also exhibit leaves with a thinner cuticle, abaxial epidermal cells with more sinuous walls, a lower occurrence of trichomes and secretory cells (laticifers), and more druse-containing idioblasts in the mesophyll. All these traits are adaptations to growing conditions that include lower light and the absence of water stress.

Conclusions

Leaf anatomical traits showed to be useful to delimit Ipomoea cavalcantei and I. marabaensis in the non-reproductive stage. For individuals of I. cavalcantei cultivated in topsoil, some features, especially of the epidermis, respond to light and water supply.

Introduction

The “cangas” are Brazilian ecosystems associated with ferruginous soils and rocky outcrops (Mota et al., 2018), with a peculiar flora that exhibits many endemic species. These plants are adapted to oligotrophic environments, capable of tolerating a set of limiting environmental conditions, such as shallow soils, water deficit, low fertility, high concentrations of oxidized iron, high temperatures and radiation, and winds (Jacobi et al., 2007; Viana et al., 2016). The cangas of the Serra dos Carajás form a mountainous complex located in the Amazon Rainforest, in Northern Brazil, with one of the largest mineral reserves on the planet. Two conservation units are found in these canga areas: the National Forest of Carajás (FLONA of Carajás), which consists of a sustainable use preservation area and covers the Serra Norte and Serra Sul, where mining activities take place; and the Campos Ferruginosos National Park (PNCF), which is an integral protection area and covers the Serra do Tarzan and Serra da Bocaina (Mota et al., 2018).

Despite the economic importance and the occurrence of unique species, studies that evaluate the adaptive strategies of canga plants are relatively scarce when compared with other Brazilian rocky outcrops. For example, different studies have been carried out with endemic plants from rocky outcrops of the Espinhaço Mountain Range (Minas Gerais and Bahia), which also grow under conditions of high radiation, thermal oscillation, and water stress. Species of Microlicia (Melatomataceae) that have grown in that environment, for example, have isobilateral leaves or leaves with a homogeneous mesophyll consisting only of palisade parenchyma (Silva, Romero & Simão, 2018). Other species, as seen in bromeliads and members of family Velloziaceae, have mechanisms to delay the water loss or store water in the leaves, including the predominance of stomata on the abaxial leaf surface, mucilaginous cells, and water-storage tissue (e.g., Silva, Versieux & Oriani, 2018, 2020; Vieira et al., 2022). For species that grow in cangas, some distinct morphological, physiological, and anatomical adaptations are expected to be selected in response to high solar radiation, water deficit, and soil conditions (Messias et al., 2013; Carmo, Campos & Jacobi, 2016; Oliveira et al., 2016). Anatomical studies with endemic canga species adapted to ferruginous soils may provide evidence of structural responses associated with this vegetation type.

Two species, Ipomoea cavalcantei D.F.Austin, popularly known as “flor de Carajás”, and I. marabaensis D.F.Austin & Secco (Convolvulaceae), were selected as study models, as they are endemic and rare, respectively (CNCFlora, 2012), and associated with ferruginous outcrops, where they stand out in the rock vegetation (Viana et al., 2016). These species differ mainly in the color and shape of the corolla–red and hypocrateriform (gamopetalous corolla, with a narrow and tubular basis and an expanding flat apical portion) in I. cavalcantei and lilac-pink and infundibuliform (gamopetalous corolla in the form of a funnel) in I. marabaensis (see Figs. 1C–1D)–but they share morphological features of the vegetative organs and habit (e.g., deciduous shrubs) (Austin & Secco, 1988; Babiychuk et al., 2017). Such similarities, together with some specimens occurring in sympatry in some areas, make it challenging to recognize the species in the field in the non-reproductive stage. Therefore, in addition to understanding the adaptive aspects associated with cangas, studies with taxonomic purposes are necessary to delimit these species better since I. marabaensis is classified as rare by Giulietti et al. (2019) and I. cavalcantei is considered critically endangered and exclusive to the Serra Norte of FLONA of Carajás. This species is also on the red list of the National Center for Flora Conservation (CNCFlora, 2012). In this sense, I. cavalcantei has been a priority for conservation (Mota et al., 2018), with several specimens rescued from areas to be mined and kept under cultivation for subsequent translocation (Viana et al., 2016). Studies that evaluate structural changes in the plant under new growth conditions can evaluate the morphological and adaptive plasticity of the species (Silva, Versieux & Oriani, 2018, 2020). Anatomical studies with vegetative organs, such as the leaf, are important for understanding adaptive responses and taxonomy issues. The main reason is that the leaf is the organ with the most significant variability of anatomical features, capable of expressing the environmental conditions (Esau, 1976; Fahn, 1982; Cutler, 1986), with some of these variations contributing to the taxonomy of related species (Metcalfe & Chalk, 1950).

Figure 1 Habit and morphology of Ipomoea cavalcantei and I. marabaensis.

(A) Canga vegetation at Serra dos Carajás. (B, E) Specimens of I. cavalcantei kept in the nursery. (C, D) Species in the natural environment: I. cavalcantei and I. marabaensis, respectively. (photographs B & E by F.C. Pinho).

This work evaluates qualitative and quantitative anatomical characters through light microscopy (LM) of the leaf blade of I. cavalcantei and I. marabaensis with taxonomic and adaptive value. We seek to answer the following questions: (A) What leaf anatomical characters contribute to delimit these species under natural conditions? (B) Do the species have anatomical convergences related to the canga environment? (C) How do I. cavalcantei leaves (rescue plants) respond structurally to cultivation conditions? The last two questions provide adaptive evidence associated with canga vegetation, which is necessary to understand how the species adapts to conditions different from those in which it was established in a natural environment.

Materials and Methods

Collection of botanical material (Figs. 1A–1E)

The material was collected in canga areas (Fig. 1A) of Serra Norte (Ipomoea cavalcantei; Fig. 1C; −50.282647/−6.030078; BHCB 157955) and Serra Sul (I. marabaensis; −62344/−929811; BHCB 130697) of FLONA de Carajás, and Serra da Bocaina of PNCF (I. marabaensis; Fig. 1D; −49.890098/−6.3133; BHCB 158287). The collection license number is SISBIO # 76784-1, issued by ICMBio.

For I. cavalcantei, many specimens occur in different mined canga areas of Serra Norte and, therefore, they were transplanted and maintained under cultivation in topsoil in a nursery (Vale SA–Carajás nursery; Figs. 1B, 1E). After acclimatization and development of new branches, leaves were collected under the cultivation conditions.

To standardize collection the expanded leaves were collected from the 4th–6th nodes (from the shoot apex) of the two species and the two growth conditions of I. cavalcantei (canga and nursery). The material was fixed in FAA 70 (37% formaldehyde, glacial acetic acid, 70% ethanol, 1:1:18 v/v; Johansen, 1940) and stored in 70% ethanol.

Anatomical study

Three individuals per species and growth condition (canga and nursery) were analyzed for leaf characterization. Leaves stored in 70% ethanol were previously observed under a stereomicroscope (Zeiss, Oberkochen, Germany, SteREO Discovery V12) to observe trichomes. Subsequently, fragments (midrib, margin, and region between margin and midrib) were obtained from the middle region of the blade. These samples were subjected to dehydration in an ethanol series (80%, 90%, 95%, 95%), followed by embedding in (2-hydroxyethyl)-methacrylate (Leica Historesin Embedding Kit, Wetzlar, Germany) (Gerrits & Smid, 1983). Cross sections (5 µm thick) were obtained with a semi-automatic rotary microtome (RM 2255; Leica, Wetzlar, Germany) and arranged on slides. Freehand longitudinal sections were additionally taken to aid in trichome characterization. All sections were stained with toluidine blue (O’Brien, Feder & McCully, 1964) and mounted with Entellan (Merck, Rahway, NJ, USA) or in distilled water.

Histochemical tests were performed with sudan III for lipids (Sass, 1951) in order to observe the cuticle, and with ferric chloride (Johansen, 1940) and ruthenium red (Chamberlain, 1932) to confirm the presence of phenolic compounds and mucilage/pectic compounds, respectively, in the epidermal cells.

The epidermis was also dissociated from the middle region of the blade using a solution of hydrogen peroxide and acetic acid for 24 h (modified from Franklin, 1945); the epidermis was stained with Safranin (modified from Bukatsch, 1972) and mounted on slides with 50% aqueous glycerin (Purvis & Colier, 1964).

All slides were analyzed, and images were taken using a light microscope (Zeiss Axio Scope.A1, Jena, Germany) equipped with a camera (Zeiss AxioCam ICc 5; Zeiss, Jena, Germany) and AxioVision software (version 4.8.3.0).

Quantitative analysis

The stomata were quantified using the epidermal printing technique (Segatto et al., 2004) on the abaxial surface (region between margin and midrib) with adhesive glue (super bonder®). It was applied to two leaves (from the fourth and sixth nodes, preserved in 70% ethanol) from two individuals each of I. marabaensis, I. cavalcantei in canga, and I. cavalcantei under cultivation, resulting in 12 leaves. These analyses were conducted on the abaxial surface, as stomata are scarce or absent on the adaxial surface of the middle region of the lamina (see Results)–physiologically hypostomatic leaves (Muir, 2015; Richardson, Brodribb & Jordan, 2017). The prints were observed under light microscopy. Images of the stomata were obtained for five fields (0.06 mm2), with a 40X objective. The following parameters were measured per field: number of stomata, polar and equatorial diameter of stomata (µm; in open stomata), and stomatal density (mm2). The number of stomata was used to calculate stomatal density (Castro, Pereira & Paiva, 2009). Stomatal functionality was obtained from the polar diameter/equatorial diameter ratio (Castro, Pereira & Paiva, 2009).

The thickness (in µm) of leaf tissues (fourth and sixth nodes) of three individuals for each condition (I. marabaensis, I. cavalcantei in canga, and I. cavalcantei under cultivation) was also measured (total number of leaves = 18). We measured the thickness of the epidermis (adaxial and abaxial surfaces), palisade parenchyma (adaxial and abaxial in I. marabaensis) spongy parenchyma, and the total mesophyll in a region between the margin and the midrib, in the middle region of the blade (we used the same slides performed for leaf characterization). The means and standard deviations of all variables were calculated.

Counts and measurements of stomata and tissue thickness were performed using ImageJ analysis software (National Institutes of Health, Bethesda, MD, USA), calibrated with a microscopic ruler photographed at the same magnifications as the micrographs. All data analyses were carried out in R environment (R Core Team, 2016). Before mean comparisons, the data were checked for normality using the Shapiro-Wilk test with the (shapiro.test function) of the R package. Then, we carried out a one-way analysis of variance with the ANOVA function followed by a subsequent Student’s t-test. We compared the species in the same environment (canga) and I. cavalcantei, which has grown naturally, and plants cultivated in nursery conditions. Figures were prepared with the package ggplot2.

Results

Both Ipomoea species studied have amphistomatic leaves, with stomata visually more abundant on the abaxial surface (Figs. 2A–2C). On the adaxial surface, stomata are scarce (Fig. 2D) or even absent at the middle region of the lamina in some specimens. Paracytic stomata (Fig. 2E) occur in all specimens studied, but in I. marabaensis the diacytic type (Fig. 2F) also occurs. The leaves of individuals of I. cavalcantei grown in a natural environment did not differ from I. marabaensis in stomatal density, functionality, and size (Figs. 3A–3D). On the other hand, the leaves of I. cavalcantei under cultivation conditions showed lower stomatal density (Fig. 3A) but higher stomatal functionality (Fig. 3B) and stomata with higher polar diameter (Fig. 3C).

Figure 2 Leaf epidermis of Ipomoea cavalcantei from nursery and canga plants, and of I. marabaensis.

Leaf epidermis of Ipomoea cavalcantei from nursery (A, D, E, G, H) and canga plants (B, J), and of I. marabaensis (C, F, I), in frontal view (A–G) and in cross (H, I) and longitudinal sections (J). (A–C) General aspects of the abaxial surface. (D) General aspect of the adaxial surface. (E, F) Details of the stomata of the paracytic and diacytic types, respectively. (G, H) Peltate glandular trichomes. (I) Capitate glandular trichomes. (J) Non-glandular trichomes. st, stomata; gt, glandular trichome; nt, non-glandular trichome. Scale bars: (A–C), (H–J): 50 μm, D: 30 μm, (E–G): 20 μm. Asterisks (*) indicate subsidiary cells.

Figure 3 Quantitative anatomy of the stomata of Ipomoea cavalcantei (nursery and canga), and I. marabaensis.

(A) Stomatal density. (B) Stomatal funcionality. (C) Polar diameter. (D) Equatorial diameter. ns, no statistically significant difference; *, **, with significant statistical difference (p < 0.05). The asterisks mean * = 0.05; ** = 0.01.

In the frontal view, the anticlinal walls of epidermal cells are more sinuous on the abaxial surface of I. cavalcantei leaves under cultivation (Fig. 2A) than the leaves from the natural environment (Fig. 2B) and I. marabaensis (Fig. 2C). On the adaxial side, the anticlinal walls are straight or curved, as illustrated for I. cavalcantei under cultivation (Fig. 2D). Glandular (Figs. 2D, 2G–2I) and non-glandular (Fig. 2J) trichomes occur on both sides of the blade. Glandular trichomes have a stalk of two cells and a multicellular (peltate trichomes; Fig. 2H) or unicellular head (capitate trichomes; Fig. 2I). The latter was observed mainly on the veins. Peltate glandular trichomes generally occur in depressions (Fig. 2H). The non-glandular trichomes comprise a basal cell, a short collar cell, and a long apical cell (Fig. 2J).

In cross-section, the blade has a single-layered epidermis with thin-walled cells in all the specimens studied (Figs. 4A–4C). The epidermal cells on the adaxial surface have a larger lumen than those on the abaxial side (Figs. 4A–4C). The thickness of the adaxial epidermis did not differ between the two species studied and the two growth conditions for I. cavalcantei (natural environment and nursery). However, the thickness of the epidermis on the abaxial surface was greater in I. marabaensis (Fig. 5B). Variations in cuticle thickness were observed, including tests for lipid detection (test with sudan III). Thus, in I. cavalcantei from the natural environment and I. marabaensis, the epidermis on the adaxial surface is covered by a thickened cuticle with ornamentation (Figs. 6A, 6E, 6G). Visually, the leaves of I. cavalcantei under cultivation exhibit a thin and smooth cuticle covering the epidermis on the adaxial surface (Figs. 6B, 6F); on the abaxial surface, the cuticle is thin in all samples studied. Similarly, we observed secretory cells (Fig. 6A–arrow) in the epidermis of all leaves; histochemical tests with ferric chloride and ruthenium red confirmed a mixed composition for phenolic compounds (Fig. 6H–arrow) and mucilage (Fig. 6I arrow), respectively. In all species the stomata occur at the same level or slightly above the other epidermal cells (Figs. 4A–4C; 6C).

Figure 4 Leaf margin of Ipomoea cavalcantei from nursery and canga plants, and of I. marabaensis.

Leaf margin of Ipomoea cavalcantei from nursery (A) and canga plants (B), and of I. marabaensis (C), in cross sections. (A–C) General aspects. Arrow, idioblast; st, stomata; vb, vascular bundle; sp, spongy parenchyma; pp, palisade parenchyma; gt, glandular trichome; *, secretory cell. Scale bars: 100 μm.

Figure 5 Quantitative anatomy of the epidermis and mesophyll of Ipomoea cavalcantei (nursery and canga), and I. marabaensis.

(A, B) Thickness of the adaxial and abaxial surfaces of the epidermis, respectively. (C, D) Thickness of the adaxial and abaxial palisade parenchyma (only in I. marabaensis), respectively. (E) Thickness of the spongy parenchyma. (F) Total mesophyll thickness. ns, no statistically significant difference; *, ***, with significant statistical difference (p < 0.05). The asterisks mean * = 0.05; *** = 0.001.

Figure 6 Details of the median region of the leaf blade of Ipomoea marabaensis, and I. cavalcantei from nursery and canga plants, in cross sections.

Details of the median region of the leaf blade of Ipomoea marabaensis (A, G), and I. cavalcantei from nursery (B, C, F, I) and canga plants (D, E, H), in cross sections. (A, B) Epidermis and mesophyll on the adaxial surface of the blade. (C) Idioblasts with druses and stoma on the abaxial surface. (D) Mesophyll with secretory cell. (E–G) Histochemical tests with Sudan III indicating cuticle thickness. (H, I) Histochemical tests with ferric chloride and ruthenium red with reaction for phenolic compounds and mucilage, respectively, in epidermal cells. Arrows, cells storing content; cu, cuticle; dr, druse; pp, palisade parenchyma; *, secretory cell. Scale bars: 50 μm.

The mesophyll is dorsiventral in I. cavalcantei both under cultivation (Fig. 4A) and under natural conditions (Fig. 4B), and isobilateral in I. marabaensis (Figs. 4C; 5C–5D); in the case of I. marabaensis, intercellular spaces connected to the stomata are observed in the abaxial palisade parenchyma (Fig. 4C). In individuals grown in the natural environment of both I. cavalcantei and I. marabaensis, the palisade parenchyma facing the adaxial surface are composed of two layers of cells (Figs. 4B–4C), with a greater thickness in I. marabaensis (Fig. 5C). Under cultivation, I. cavalcantei has a palisade parenchyma changing from one to two layers of cells (Fig. 4A); the statistical analysis revealed that its thickness is not greater in relation to that of individuals from natural environment (Fig. 5C). The spongy parenchyma in all samples studied is composed of three to five layers of cells (Figs. 4A–4C) and its thickness did not show significant variation between species and growth conditions (Fig. 5E). Total mesophyll thickness is significantly greater in I. marabaensis (Fig. 5F).

Idioblasts with druses are commonly observed among the mesophyll cells (Fig. 6C), especially in the nursery plants. In I. marabaensis, these idioblasts often occur on the adaxial surface, between the palisade cells (Fig. 4C–arrow). Secretory cells are dispersed in the mesophyll of all samples (Fig. 6D), but they are observed more frequently in the leaves of I. cavalcantei from the natural environment. The vascular bundles are collateral, with a rounded shape (Figs. 4A–4C), and visually occur in greater quantity in I. marabaensis (Fig. 4C).

In the cross section, the midrib is quite prominent on the abaxial surface, with a flat outline (Figs. 7A–7C); on the adaxial surface of both species, a small protuberance may be present (Figs. 7A, 7E). The presence or absence of this feature varies in the leaves of a single individual. On the other hand, in all leaves of I. marabaensis, there are two conspicuous lateral protuberances on the abaxial surface of the midrib (Fig. 7C). The epidermal cells are rounded to rectangular on the adaxial surface (Figs. 7E, 7F) and rounded on the abaxial surface (Fig. 7H) in both species studied. Stomata eventually occur in the midrib region, on the adaxial surface of the leaf (Fig. 7F). Glandular and non-glandular trichomes are visually abundant on the abaxial surface of the midrib, mainly in plants from natural environments (Figs. 7B, 7C), being sparse in nursery plants (Fig. 7A). Still in the midrib region, the mesophyll is composed of rounded parenchymatous cells and some layers of collenchyma just below the epidermis (Figs. 7E, 7F, 7H). The arc-shaped vascular bundle is bicollateral (Figs. 7A–7D). Druse-containing idioblasts occur around and among the cells of the vascular bundle (Fig. 7G). These idioblasts are apparently more frequent in the leaves of I. cavalcantei under cultivation. Secretory cells occur in the parenchyma, around the vascular bundle, in all samples analyzed (Fig. 7H).

Figure 7 Midrib of Ipomoea cavalcantei from nursery and canga plants, and of I. marabaensis, in cross sections.

Midrib of Ipomoea cavalcantei from nursery (A, D–F) and canga plants (B, H), and of I. marabaensis (C, G), in cross sections. (A–C) General aspects. (D) Detail showing the central vascular bundle. (E, F) Details on the adaxial surface. (G) Detail of the vascular bundle showing idioblasts with druses. (H) Detail showing the secretory cells. Arrow, protuberance on the adaxial surface; co, collenchyma; dr, druse; st, stoma; ph, phloem; tr, trichome; xy, xylem; *, secretory cells. Scale bars: (A–C): 200 μm, (D): 100 μm, (E–H): 50 μm.

Discussion

Species of Ipomoea and other members of family share several anatomical features, such as paracytic stomata, glandular trichomes, an arc-shaped bicollateral vascular bundle in the midrib, druse-containing idioblasts, and latex-producing cells (Metcalfe & Chalk, 1950; Fahn, 1979; Mauseth, 1988). All these characteristics were observed in the species studied. Our results showed that the two species are quite similar anatomically. It may be due to genetic factors, which modulate the leaf structure of the genus, as well as the fact that they develop under similar environmental pressures (grow on rocky and ferruginous soils under xeric pressures). Given this, it is worth highlighting the usefulness of anatomical characters for taxonomy and those that are adaptive, in the latter case indicating convergences or changes in response to the environment.

Leaf structural characters with potential for species delimitation

Considering the characters with taxonomic importance, we highlight the type of mesophyll (dorsiventral in I. cavalcantei vs. isobilateral in I. marabaensis) and the presence (I. marabaensis) or absence (I. cavalcantei) of lateral protuberances on the abaxial surface of the midrib.

The dorsiventral mesophyll observed in I. cavalcantei is considered the most common type in eudicot leaves (Metcalfe & Chalk, 1950) and also commonly found in many Ipomoea species (e.g., Procópio et al., 2003; Martins et al., 2012; Tayade & Patil, 2012; Santos & Nurit-Silva, 2015; Porwal et al., 2015; Babu, Dharishini & Austin, 2018; Prasanth, Aleykutty & Harindran, 2018; Santos et al., 2020; Ekeke, Nichodemus & Ogazie, 2021; Santos et al., 2023). On the other hand, the isobilateral mesophyll is less frequent, observed in a few species of the genus, such as I. imperati (Vahl) Griseb., I. pes-caprae (L.) R.Br. (Arruda, Viglio & Barros, 2009), I. burchellii Meisn. (Santos et al., 2020), and I. marabaensis (studied here).

The shape of the midrib of the Ipomoea species studied was similar, with a flat abaxial surface but with lateral protuberances in I. marabaensis. Such protuberances have also been observed in I. maranhensis (Santos et al., 2020). It is worth noting that a small protuberance on the adaxial surface of the midrib may be present in Ipomoea leaves, as observed in the species studied here and in I. imperati (Kuster et al., 2016); our data indicated this character as an intraspecific variation, and therefore not useful for species delimitation. However, other studies have indicated taxonomic value for this structure. Santos et al. (2020) described a conspicuous protuberance on the midrib of I. maranhensis, differentiating this species from I. burchelli. Ekeke, Nichodemus & Ogazie (2021), studying I. coccinea, observed a long protuberance on the adaxial surface of the midrib and on the petiole, which supports taxonomic delimitation. Other studies indicate that this adaxial protuberance is proeminent and collenchymatous (Babu, Dharishini & Austin, 2018; Santos et al., 2023), which may give the triangular shape to the midrib of I. pes-tigridis, together with the flat contour of its abaxial surface (Babu, Dharishini & Austin, 2018).

In a comparative analysis, epidermal appendages can be discussed taxonomically. As an example, there are the paracytic-type stomata that are common for many Convolvulaceae species, including I. cavalcantei and I. marabaensis (Metcalfe & Chalk, 1950; Tayade & Patil, 2011). However, I. marabaensis still has some diacytic stomata, which helps in its delimitation. It is worth noting that Ipomoea species may have a combination of one or more stomata types, being denominated heterostomatic (Dickson, 2000). In this context, the use of different types of stomata has already been used to describe many species (see Procópio et al., 2003; Folorunso, 2013; Essiett & Okono, 2014; Porwal et al., 2015; Santos & Nurit-Silva, 2015; Abba, Abdullahi & Yuguda, 2018; Bolarinwa, Oyebanji & Olowokudejo, 2018; Noraini et al., 2021).Regarding distribution, stomata on both sides of the blade are common for the family (Metcalfe & Chalk, 1950; Tayade & Patil, 2011), although their presence on the adaxial surface may be inconspicuous. Hypostomatic species (e.g., I. leari; Porwal et al., 2015) also occur. Therefore, this character may be helpful in the identification and delimitation of Ipomoea species.

Our data indicate that the leaves provide valuable characters for the taxonomy of two Ipomoea sister species, facilitating their identification in the non-reproductive stage. Comparing our results with those of other studies involving Ipomoea species, we concluded that the main characters of taxonomic importance at species level are as follows: the type and distribution of stomata, the type of mesophyll, and the structure of the midrib. We also reinforce the need for detailed anatomical studies for a better interpretation of Ipomoea structures. Non-glandular trichomes, for example, which were described as simple and unicellular structures (e.g., Essiett & Okono, 2014; Porwal et al., 2015; Abba, Abdullahi & Yuguda, 2018; Babu, Dharishini & Austin, 2018; Bolarinwa, Oyebanji & Olowokudejo, 2018; Noraini et al., 2021) are in fact pluricellular. Herein, we describe these trichomes with a basal cell, a collar cell, and an elongated apical cell. Thus, the non-glandular trichomes of Ipomoea may be simple (i.e., unbranched) but not unicellular.

Convergences and leaf functional traits related to growth conditions

The specimens from natural environments show thicker cuticles on the leaves. It is evidence of mechanical resistance against dehydration in xeric conditions (Martins, Machado & Alves, 2008; Pita, Menezes & Prado, 2006) and represents an adaptive strategy in environments such as cangas, where the possibility of water loss through evapotranspiration increases. In this sense, the long non-glandular trichomes are important structures in adapting to xeric environments, also reducing the leaf transpiration rate (Abba, Abdullahi & Yuguda, 2018), in addition to regulating the temperature in warmer periods and low water availability (Ehleringer & Mooney, 1978). On the other hand, cultivated plants of I. cavalcantei have thin cuticles and fewer trichomes, as they are subjected to a partially controlled environment, with the availability of water, nutrients, and less radiation.

Regarding the cell contour of epidermis, in frontal view, a convergence is observed. Leaves from individuals under natural conditions exhibit the same epidermal cell contour on the adaxial (straight walls) and abaxial (slightly sinuous walls) surfaces of the blade. This condition, however, changed in cultivated plants, reinforcing the selectivity imposed by the cangas. Under cultivation conditions the anticlinal cell walls are more sinuous. Thus, the shape of the epidermal cells is adjustable, responding to selective environmental pressures, especially regarding water availability and solar radiation. This may explain the diversity of shapes (e.g., straight, wavy, curved, slightly sinuous, and very sinuous) of the epidermal cells of Ipomoea (Procópio et al., 2003; Monqueiro et al., 2004; Arruda, Viglio & Barros, 2009; Bolarinwa, Oyebanji & Olowokudejo, 2018) and other members of Convolvulaceae (Metcalfe & Chalk, 1950).

Water availability and solar radiation are pressures that also alter the stomatal density, which is more significant in individuals from natural environments. In the leaves of I. cavalcantei under cultivation conditions, the lower stomatal density should reduce the photosynthetic rate due to its ability to acclimate to mesic conditions. According to Boeger et al. (2006), plants under lower radiation (e.g., shaded plants) generally have lower stomatal density than sunny plants, as the mild microclimate guarantees lower leaf temperatures and, consequently, a lower rate of excessive transpiration. However, the cultivated leaves have larger stomata (larger polar diameter), which would compensate for the lower density and greater stomatal functionality. Such attributes indicate efficiency in controlling the opening and closing of stomata under high water availability and low radiation in nursery conditions. Despite the amphistomatic leaves of the species studied, the stomata are scarce on the adaxial surface, indicating that these leaves physiologically act as hypostomatic (i.e., low ratio of the frequency of adaxial to abaxial stomata; see Muir, 2015; Richardson, Brodribb & Jordan, 2017). Plants with hypostomatic leaves, including those with a low adaxial/abaxial stomatal ratio, can be found in various light environments (Mott, Gibson & O’Leary, 1982). This xeromorphic condition occurs in species exposed to high solar radiation received on the adaxial leaf surface (e.g., Silva, Romero & Simão, 2018; Silva, Versieux & Oriani, 2018, 2020; Lima et al., 2022; Vieira et al., 2022), with the predominance of stomata on the abaxial surface as a structural convergence.

Phenolic compounds in the epidermis constitute another adaptive response against radiation. These compounds occur in xerophytic plants, reducing the passage of light into cells and protecting the mesophyll against excess solar radiation (Oliver et al., 2020). Furthermore, according to Kaufman et al. (1998), phenolic compounds play an important ecological role in defense against herbivory and are directly related to antioxidant activity. Antioxidant compounds are essential for maintaining plant balance, sequestering excess free radicals during the metabolic process, and preventing diseases related to oxidative stress (Michalak, 2006). Since they also store mucilage, these idioblasts still contribute to water retention and delay of water stress (Vieira et al., 2022) in the epidermis.

Considering the mesophyll, regardless of whether it is dorsiventral or isobilateral, the palisade parenchyma appears to be well-developed, especially in I. marabaensis. In this species, the elongation of palisade cells in combination with their disposition on both sides of the blade contributes to the greater thickness of the mesophyll, as also reported for other plants with isobilateral mesophyll (e.g., Silva, Romero & Simão, 2018; Lima et al., 2022). Isobilateral mesophyll differentiation occurs early in plants under high radiation conditions (Silva & Oliveira, 2014; Silva et al., 2019; Lima et al., 2022).

Idioblasts with druse-type calcium oxalate crystals occur in the leaves of the studied species, mainly in I. cavalcantei under cultivation. The abundance and even the size of these structures may be related to the age of the organ and the chemical characteristics of the substrate to which the plants are subjected (Silva, Versieux & Oriani, 2018); the latter may explain, in part, the differences observed for I. cavalcantei leaves. Some studies indicate that crystals can optimize light distribution in chlorenchyma (Larcher & Boeger, 2006; Silva, Versieux & Oriani, 2018), which would be an advantage for I. cavalcantei under cultivation conditions. Secretory cells are visually more numerous in I. cavalcantei under natural conditions, a characteristic that may have been selected due to many interactions with different types of herbivores and visiting insects (K. R. Silva, L. E. N Costa, A. C. G. Costa, C. S. Carvalho, V. C. Tavares, M. T. C Watanabe, 2024, in preparation). These secretory cells are described in the literature as latex-producing cells (Metcalfe & Chalk, 1950; Fahn, 1979; Mauseth, 1988). Fahn (1979) attributes an ecological role to the latex stored in these cells, which can heal injuries and protect plants against attacks by herbivores and microorganisms. It may confer an advantage over other species co-occurring in canga areas.

Conclusions

Leaf anatomical studies of Ipomoea are still scarce, considering its high number of species, which occur in different vegetation types. Thus, this study contributes to the knowledge of the genus, indicating characters of taxonomic value. The main anatomical characters for delimitation of the two species studied here are the type of mesophyll and the presence/absence of protuberances on the abaxial surface of the midrib. Other characters such as cuticle thickness, shape of epidermal cells, and stomatal density are related to adaptive strategies imposed by the xeric conditions of the cangas, especially water stress and high solar radiation. In this sense, the study of I. cavalcantei under cultivation conditions was essential to attest to the anatomical plasticity of the species, indicating responses to support the translocation of specimens currently occurring in mining areas. Our data can support future studies with other vegetative organs, such as roots and stems, which are still necessary for a better understanding of taxonomic and biological aspects of the species.

Supplemental Information

Supplemental Information 1 Quantitative data on stomata density and mesophyll thickness measurements in the Ipomoea cavalcantei and I. marabaensis in different conditions (natural canga areas and propagated in topsoil).

The authors thank Cesar Carvalho Neto, Vera Lúcia Imperatriz-Fonseca, Valéria Tavares and Guilherme Oliveira, for logistical support. We also would like to thank Fabia Cavalcante Pinho for logistical support and photography.

Additional Information and Declarations

Competing Interests

Author Contributions

Data Availability

The authors declare that they have no competing interests.

Joyce C. R. Silva conceived and designed the experiments, performed the experiments, analyzed the data, prepared figures and/or tables, authored or reviewed drafts of the article, and approved the final draft.

Kleber R. Silva conceived and designed the experiments, performed the experiments, analyzed the data, prepared figures and/or tables, authored or reviewed drafts of the article, and approved the final draft.

Cecilio F. Caldeira analyzed the data, prepared figures and/or tables, authored or reviewed drafts of the article, and approved the final draft.

Aline Oriani conceived and designed the experiments, analyzed the data, authored or reviewed drafts of the article, and approved the final draft.

Mauricio T. C. Watanabe conceived and designed the experiments, analyzed the data, authored or reviewed drafts of the article, and approved the final draft.

The following information was supplied regarding data availability:

The raw data are available in the Supplemental File.

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
