# Peer review of "Comparative leaf anatomy of two species of Ipomoea L. (Convolvulaceae): taxonomic importance and adaptations to xeric conditions of the cangas"

_PeerJ, doi:10.7717/peerj.18599_

## Round 0.1 · original submission · Major Revisions

Apart from the editorial corrections suggested by both reviewers (especially reviewer 2), there are some issues with the interpretation of results and the conclusions arising from them, that need to be addressed in the revision. Reviewer 1 highlights the need for the Discussion to be more focused and concise while Reviewer 2 emphasizes the fact that this study is based on only two species so that conclusions may not be applicable to the whole family.

Therefore, the required revisions include both revisions in content and language.
The authors are advised to address all these points raised by the two reviewers in on a point-by-point basis.

Reviewer 1 ·

Basic reporting

The authors conducted a comparative study of two Ipomoea species. The micrographs are all clear and the quantitative analyses were made in a proper way.

Experimental design

Although the aim of this study was to find out the key characters for species identification of these species in the vegetative phase, the anatomical and morphological characters identified in this study may not be robust. For example, the authors did not check the growth environment effects on the two protuberances on the midrib on the abaxial side in I. marabaensis. Judging from the photos in Figure 1, the length /width ratios of the leaves are very different between the two species. Can this be a useful key?

Validity of the findings

How did the authors distinguish palisade tissues from spongy tissues? The leaves with the palisade tissues on both sides tend to obliquely of vertically oriented?

Additional comments

The authors use many anatomical/morphological terms. Many readers may not be familiar with such terms. Some explanations may be added to very specific terms.

In the first paragraph of Introduction, many Brazilian vegetations are described. A vegetation map may be very helpful.

Discussion is too long. Although there are two subtitles, topics are too numerous. The effects of growth environment, which were analysed in one species, agree with the general trends found between the plant materials grown under mesic conditions and those under xeric conditions. Such general features may be more concisely argued.

·

Basic reporting

Improvements are required on the language. Minor changes in formatting, figure legends, figures.


There is a need to improve grammar and overall sentence construction in certain places as given below. This is required for easy comprehension.
Line 25 - Between to be replaced by of ‘…..characters between the species were described’
Line 44 – re-word this section as it is unclear – ‘are vegetation-like associated with..’
Line 58-62 – Break into two sentences from line 60. Can start with, ‘For example, different studies have been…’
Line 63,64 – Sentence is unclear. ‘…..isobilateral leaves or a fully developed mesophyll into palisade parenchyma.’
Line 83,84 – remove ‘is’ and the comma after Carajás ‘…. I. cavalcantei is exclusive to the Serra Norte of FLONA of Carajás, is considered critically endangered..’
Line 93 - re-word sentence – ‘may have a taxonomic value’
Line 110 – Connection unclear – ‘Collect license number SISBIO # 76784-1.’
Line 111-113 – unclear
Line 138 – correct as, ‘… and images were taken using a light microscope’
Line 153 - correct as, ‘…used to calculate stomatal density’
Line 195 - correct as, ‘The thickness of the adaxial epidermis did not differ..’
Line 236 – avoid use of the word ‘seem’
Line 242 – corrects as ‘..other member of family..’
Line 244 – cells
Line 272 - corrects as '..and therefore not useful for species…’
Line 273 – the sentence on taxonomic value of the protuberance unclear
Line 277 – replace the words ‘is very evident’
Line 218 – corrects as, ‘arc-shaped mid rib..’
Line 313 - corrects as. ‘at species level’
Line 315 – members of family..
Line 333 – correct as, ‘…water loss through evapotranspiration high’
Line 348 – remove ‘the’
Line 362 – reword – ‘Still considering the epidermis..’
Line 390 – replace ‘luminosity’
Line 397 – replace ‘after’ as it makes the sentence unclear
Line 409 – replace ‘luminosity’
Line 424, 425 – reword as it is unclear
Line 456 – photography


The use of hyphens to connect ideas and sentences continuously is discouraged as it does not lend to a smooth reading of the paper nor to grammatical correctness.
Line 86-89
Lines 158 – 162 (particularly a hyphen and a comma seen in line 160)
Line 283-284
Line 297-298
Line 334-337

Further specific comments,
Line 39 - adaptive responses to what?
Line 65 – Should be written as members of family Velloziaceae
Line 69 – high solar radiation
Line 152 - Ostiole is not a commonly used term for stomatal studies
Line 317 – use of ‘our data’
Line 325-327 – this would be only for these two species?
Line 406,407– unclear - considerable variation in ‘what’ as it is not mentioned.
Line 440 – it is only for the two species studied
Line 443 – high solar radiation


Related to formatting
Line 72 and 73 – add space before Austin
Line 131 -133 – uniformity in terms of capitalization of reagent names
Line 352 - CO2 abbreviation only is used
Line 366 - CO2 given as CO2

Related to figures
• Figure 3, 5 – significance use of asterisk - *, ** and *** - the difference between these are not mentioned.
• Figure 6 – The large arrowheads to show the cuticle is distracting and hence it is better to use lettering instead.
• Figure 7 – recommended to use an arrow instead of an arrowhead for the protuberance

Experimental design

It would have been good to have a higher replication (more individuals)
The use of stomatal index is encouraged over stomatal density.

The manuscript focuses on the anatomical aspects of two species belonging to family Convolvulaceae. The detailed work on the leaf anatomy and well taken anatomical sections are commended.

Validity of the findings

Though histochemical tests are mentioned in the methodology their use in the results is somewhat unclear.
The conclusion speaks of the entire genus though the study was on two species of the genus.
Line 447-449 The latter part of the conclusion needs to be improved as it does not provide an overview of the study but instead speaks solely of further studies.

Additional comments

References related

Many of the references are somewhat dated.

· The reference list and in text citations need to be checked.

Some references are not in the body of the paper. Eg:

o Lines 463

o Line 521

o Line 528

o Line 636

o Line 639



· Page numbers not given

o 507 Fahn A. 1979. Secretory Tissues in Plants. London: Academic Press Inc

o 508 Fahn A. 1990. Plant anatomy. Pergamon Press, Oxford



· In line 82 Giulietti et al. (2009) should be 2019



· Line 495 Carmo FF, Campos IC, Jacobi CM. 2016.

Not in alphabetical order in the reference list as it comes after letter ‘D’

---

## Round 0.2 · Minor Revisions

Please address the minor revisions suggested by the two reviewers. As Reviewer 1 has left the option of addressing their comments to the discretion of the authors, if the authors decide not to address, please provide necessary justifications for doing so.

Reviewer 1 ·

Basic reporting

The revised version reads well.

Experimental design

See the comment in 4.

Validity of the findings

See the comments in 4.

Additional comments

The revised version reads well. However, this reviewer has still a few comments.

1. To one of the previous comments, the authors responded as follows:

The authors use anatomical and morphological terms widely recognized and used in plant anatomy. Such terms are essential to describe the structures and characteristics of species accurately. Standardizing these terminologies allows for the reproducibility of results since experts in the field widely accept them.

Although this may be right, but addition of a few words for each term is enough for smooth reading for the readers who are not familiar with these terms. This reviewer would like ask the authors to be more kind to such readers. For example, the terms below are not necessarily so familiar. Illustrations as the supplementary data are very helpful.

hypocrateriform, infuncifuliform, paracytic, actioncytic, anisocytic, anomocytic, barachyparacytic, cyclocytic, diacytic, laterocytic and staurocytic.

For stomatal complexes of the present two species, it is necessary to clearly indicate the cells which are identified as the subsidiary cells by the author in Fig. 2. This reviewer feels that the stomatal complexes shown in Fig. 2 are not necessarily all paracytic.

2. Stomatal functionality: If the replicas are taken from live samples, the stomatal pore area is roughly calculated using the polar and equatorial diameters. Then, it may be called ‘stomatal functionality.’ But, this reviewer does not understand how the ratio tells functionality. This reviewer also wonders about the data obtained with the fixed samples. Because these axes (in particular, polar axis) are still good indicators of stomatal size, these data are valuable. However, the use of the term ‘functionality’ should be reconsidered. As another reviewer pointed out and as the authors know very well, the stomatal index is also a very useful parameter. This is because the absolute stomatal size and cell density are usually inversely correlated. Because the photos in Fig. 2 are clear enough, why not show the data of stomatal index as well? Then, the effects of growth conditions can be argued easily.

3. For the readers who are not familiar with the Brazilian states, the 1st paragraph in the Introduction conveys little information.

Minor comments

Plant description: Are these plants herbaceous or shrubs? Are these evergreen or deciduous?

L. 66: store (water?) and/or delay water loss.

L. 154: Why does the ratio indicate the stomatal ‘functionality.’ This reviewer cannot agree with this idea.

L. 330: The stomatal index data support the argument, here.

·

Basic reporting

No comment. The suggested changes have been adhered to/justified if not.

Experimental design

The justification with regard to replication is acceptable.

Validity of the findings

The use of histochemical tests is still unclear in the overall study.

Additional comments

The significance levels have been added to the figure legends. They are given as 0,05 etc. Should be 0.05?

---

## Round 0.3 · accepted · Accept

The authors have addressed all comments and issues raised by the two reviewers satisfactorily. Hence, the manuscript can be accepted.